# Pricing and Coordination Strategies of Dual Channels Considering Consumers' Channel Preferences

**Rufeng Wang** [1] , **Siqi Wang** [1] **and Shuli Yan** [2],*

1   School of Management, Henan University of Science and Technology, Luoyang 471000, China;
    wangrus992@tju.edu.cn (R.W.); 200320190892@stu.haust.edu.cn (S.W.)
2   School of Management Science and Engineering, Nanjing University of Information Science and Technology,
    Nanjing 210044, China
*   Correspondence: yshuli@126.com; Tel.: +86-177-681-28192

**Abstract:** With the rapid development of electronic commerce, consumers can freely buy the same product from a manufacturers' Internet channel or a resellers' physical channel. Based on the consumers' channel preferences, this article classifies consumers into three types and investigates the price decision in a dual-channel supply chain using a Stackelberg game, which assumes that the manufacturer, as the game leader, first sets the wholesale price, then the reseller decides the retail price, according to the wholesale price. Furthermore, some numerical experiments are developed to investigate the impact of consumer acceptance, the degree of customer loyalty, and the proportion of identical shoppers on prices and profits. The results show that whether both the retail price and the wholesale price rise or fall depends on a combination of the cost of the physical channel and the Internet shopper's acceptance of the Internet channel. The reseller's profit is always lower than the manufacturer's profit. The reseller's profit is lower and the manufacturer's profit is higher, compared with that of a traditional single channel supply chain. The numerical experiments showed that when an Internet shopper's acceptance of an Internet channel is lower, the wholesale price and retail price in the dual channels will increase with an increase of the degree of customer loyalty (the proportion of identical shoppers). The reseller's profit (the manufacturer's profit) will reduce (rise) with the augmentation of the Internet shopper's acceptance of an Internet channel. Finally, we design a revenue-sharing contract that can coordinate the supply chain and implement a win–win strategy for all partners. This work makes some contributions to the research area of coordination in dual-channel supply chains.

**Keywords:** dual-channel supply chain; pricing strategy; coordination; channel preference; Stackelberg game



## 1. Introduction

With the rapid development of electronic commerce, manufacturers in more and more industries have introduced Internet channels, in addition to their traditional physical retail channels. For example, computer manufacturers (e.g., Apple, IBM, and Cisco), cosmetics manufacturers (e.g., Estee Lauder), beverage and food manufacturers (e.g., Coca Cola), sports manufacturers (e.g., Nike), and electronics producers (e.g., palmOne, Samsung, and Sony) have adopted a dual-channel supply chain consisting of an Internet channel and a traditional physical retail channel [1]. Facing the emergence of Internet channels, resellers have experienced fiercer competition from manufacturers' Internet channels and complain that the manufacturers encroach on their market. Thus, channel conflict may occur in dual-channel supply chains.

In the era of big data, the Internet is developing rapidly, and online shopping is more convenient. Consumers' purchasing behavior will change under a dual-channel environment. Many researchers have discussed this issue. Balakrishnan et al. [2] investigated the browse-and-switch behavior of consumers; i.e., consumers first visited the traditional

retail store and then switched to an Internet channel to buy the product, because the online price of the product was less than that in the retail store. Indeed, it is possible for some consumers to pay a higher price on visiting a physical channel than just clicking online. Based on the practical situation, Hsiao and Chen [3] classified consumers into two types: the grocery shopper, who had a lower willingness to buy on an Internet channel, and the Internet shopper, who had a higher willingness to buy on an Internet channel.

Nevertheless, there must be some people who have the same willingness to use on Internet channels and physical channels, especially when they can obtain the product at the same price wherever they buy it. For these consumers, it does not matter waiting a while to obtain a product if they purchase it online, or they do not mind paying a higher price on visiting the physical channel. Therefore, it may be of the same value for them to purchase the product online or offline. In a word, when the consumer can purchase the same product from an Internet channel and a physical channel, they prefer the channel which is suitable to their requirements in a dual-channel supply chain [4]. In this paper, we categorize consumers into three types (see Section 3) and discuss the pricing strategy of dual-channel supply chains.

This raises three natural questions: (1) how the supply chain decides the price based on the three types of consumers, who have different channel preferences? (2) how the proportions affect the price decision?; and (3) whether a contract can be designed to coordinate a dual-channel supply chain? Motived by these questions, several important issues arising from the pricing strategy are addressed: (1) pricing decisions based on three types of consumers, who have different channel preferences; (2) the impact of consumer acceptance of the Internet, the degree of customer loyalty to the physical channel, and the proportion of identical shoppers on the pricing decisions; (3) coordination with a revenue-sharing contract. Our main contribution is to solve these problems. Using Stackelberg games, the price decision is investigated considering three types of consumer in dual-channel supply chains, and some numerical experiments are presented to investigate the impact of consumer acceptance, the degree of customer loyalty, the proportion of identical shoppers on prices and profits. Additionally, a revenue-sharing contract is designed that can coordinate decentralized dual channels.

## 2. Literature Review

The literature concerning channel competition in supply chains has been widely studied [5–7], especially between an e-channel and a traditional retail channel [8–10]. As for dual-channel supply chains, one stream in prior literature has focused on the issue of whether the manufacturer adopts a consistent pricing strategy in a dual-channel supply chains, while a second stream involves the coordination of the dual-channel supply chain. The last stream is about consumer heterogeneity with the valuation of products.

First, a lot of research about channel competition has tended to focus on pricing strategies. In dual channels, Cai et al. [11] stated that an equal-pricing strategy could bring down the levels of channel conflict. Therefore, many researchers used equal-pricing strategy in dual-channels in the following research. For example, Cattani et al. [9] showed that when an equal-pricing strategy was applied, both the manufacturer and the reseller obtained higher profits. An equal-pricing strategy was the optimal policy adopted by the manufacturer in a dual-channel supply chain [12]. Li et al. [6] adopted a consistent pricing strategy in a dual-channel supply chain, in which they solved a Nash bargaining problem, where the manufacturer used bargaining strategies for prices and the order quantity with the retailer when the retailer had equal bargaining power. Differently from the Nash bargaining game in Li et al. [6], we established a Stackelberg game model, where the manufacturer is the game leader and the retailer is the game follower. The Stackelberg game was proposed in 1952 by Heinrich Von Stackelberg, who was a German economist [13]. As is well known, in a Stackelberg game one player acts first and the other player acts second. Many scholars have used Stackelberg games to investigate price strategies in

supply chain management [14–16]. Similarly to Li et al. [6], we adopted consistent pricing in a dual-channel supply chain.

As is well known, if a supply chain can be coordinated, the partners benefit from the coordination [17]. However, how is the supply chain coordinated? Geng and Mallik [18] put forward a verse revenue sharing contract, in which the manufacturer gave a proportion of the profit of the direct channel to the retailer. They showed that this contract could help coordinate a dual-channel supply chain. Cai et al. [11] found that a simple price discount contract could effectively improve both the manufacturer's and the retailer's performance, but could not coordinate the dual-channel perfectly. Chiang [19] proposed a contract that coordinated a dual-channel supply chain, under which the manufacturer not only transferred a proportion of the profit of the direct channel to the retailer, but the two parties also preferred to share the total inventory holding cost by transferring payments. Xu et al. [20] put forward a two-way revenue sharing contract that coordinated a dual-channel supply chain with risk-averseness. The agency theory was used to test the functioning of asymmetric information in a coordinated supply chain in [21]. Furthermore, Zhang et al. [22] made use of a contract related to the wholesale price, the direct price, and a lump sum fee, to coordinate a dual-channel supply chain under, not only the case of demand disruptions, but also the case of production cost disruptions. Basiri and Heydari [23] investigated the impact of consumer environmental awareness and a product's green quality on the coordination of a substitutable supply chain. Biswas et al. [24] found that a sustainable risk-neutral supply chain could be coordinated, and in which a greening effort dependent demand was faced. Li et al [25] designed a coordination contract for a dual-channel supply chain, which considered a socially responsible manufacturer. Although Boyaci [26] showed that a revenue sharing contract could not coordinate a dual-channel supply chain, in this paper a revenue sharing contract can coordinate a dual-channel supply chain when the manufacturer adopts an equal-pricing strategy.

In the studies of dual-channel supply chains, there are many articles that investigate consumer heterogeneity regarding the valuation of the products [27–30]. Luo and Sun [31] examined how a manufacturer might use product design to influence the retailer's outlet designation decision in a dual-channel supply chain, in which consumer heterogeneity was considered. Zhang and Wang [32] focused on the pricing and the level of product greenness in a supply chain where the consumers had different greenness preferences. Without a loss of generality, we adopt similar assumptions to Chiang et al [8] and Hsiao and Chen [3], in which the total market size is constant, the consumers' value follows a uniform distribution, and the demand functions are induced. Differently from in their research, the consumers are classified into three segments, according to their different channel preferences. The pricing decisions of a decentralized traditional single channel and a decentralized dual-channel supply chain are solved and are compared and analyzed. Finally, revenue-sharing contracts for the coordination of the dual-channel supply chain are designed. In our work, the revenue-sharing contract, not only can coordinate the decentralized dual-channel supply chain, but also implement a win–win strategy for all supply chain partners.

The paper is organized as follows: In the next section, the problem is described briefly and some notations are given. Section 4 establishes the model and analyzes the solutions. In Section 5, several numerical experiments are provided. Finally, in Section 6, we give some managerial implications and our conclusion. Proofs of the propositions and corollaries are shown in the Appendices A–E.

## 3. Description of the Problem

In this study, we investigate a supply chain that consists of a monopoly manufacturer, an independent reseller, and heterogeneous consumers regarding the value of the product. The consumer can obtain the product at the same price from the reseller's physical channel and from the Internet channel of the manufacturer. The consumers are categorized into three types: the grocery shoppers, who obtain a higher value in the reseller's physical

channel (the first segment); the Internet shoppers, who obtain a higher value from the Internet channel (the second segment); and the identical shoppers, who obtain an equal value, whether from the physical channel or the Internet channel (the third segment). All parties have common information about the supply chain and are risk-neutral and seek to maximize their expected profits. The following are some important assumptions:

(1) The manufacturer charges the wholesale price $w$ from the reseller and the consumer can get the product at the price $p$ from the physical channel or Internet channel.

(2) Every consumer is willing to buy one, and only one, unit of product. Without losing generality, the market size is normalized to 1 [3,8]. We use $s(1-t)$, $(1-s)(1-t)$, and $t(0 < t < 1)$ to denote the proportions of consumers in the first segments, the second segments, and the third segments, where $s(0 < s < 1)$ is the degree of customer loyalty to the physical channel.

(3) For the sake of simplicity, the marginal production cost is normalized to 0. The cost of the physical channel and the cost of the Internet channel are assumed to be $c$ ($c > 0$) and 0, respectively.

(4) Each consumer's valuation obtained from the reseller is $v$ ($v \sim U[0,1]$).

Now, the grocery shopper will obtain $\alpha v$ ($0 < \alpha < 1$) from the Internet channel, the Internet shopper will obtain $\beta v$ ($\beta > 1$) from the Internet channel, and the identical shopper will obtain $v$ from the Internet channel. From what we discussed above, the consumer who is the grocery shopper obtains a net utility $u_{1r} = v - p$ from the reseller and obtains a net utility $u_{1d} = \alpha v - p$ from the Internet channel. As $0 < \alpha < 1$, it is not difficult to obtain $u_{1r} > u_{1d}$, and the grocery shopper will only purchase the product from the reseller. Similarly, the Internet shopper will only purchase the product from the Internet channel ($u_{2d} = \beta v - p \geq u_{2r} = v - p$). However, the identical shopper obtains an equal net utility from both the physical channel and the Internet channel, $u_{3d} = u_{3r} = v - p$. Without loss of generality, we can assume that 50% of the third segment will purchase the products from the reseller and 50% of the third segment will purchase the products from the Internet channel. Hence, we can obtain the demand function of the reseller $Q_r = s(1-t)(1-p) + t(1-p)/2$ and the demand function of the Internet channel $Q_d = (1-s)(1-t)(1-p/\beta) + t(1-p)/2$.

## 4. Models

### 4.1. The Traditional Single Channel Supply Chain: A Benchmark

First, we start with the traditional single channel supply chain model, which is formulated with a manufacturer, a traditional reseller, and a continuum of consumers with heterogeneous preferences, in which the reseller operates a physical channel. As the Stackelberg game leader, the manufacturer first decides the wholesale price, and then the reseller determines the retail price. The consumers whose valuation satisfies $v - p \geq 0$ will purchase the product. The demand function is $Q = 1 - p$. The game sequence is as follows (as shown in Figure 1).

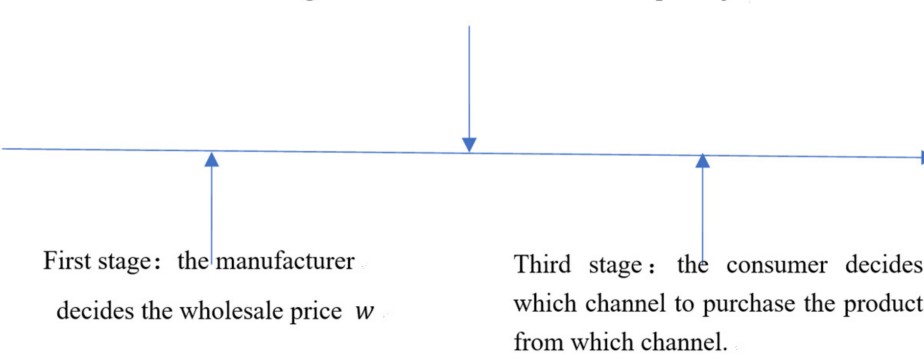

**Figure 1.** The game sequence.

By backward induction, the reseller's decision problem in the second stage is

$$\max_{p} \pi_{Tr} = (p - w - c)(1 - p). \tag{1}$$

Solving (1), we have

$$p_T^* = (1 + c + w)/2 \tag{2}$$

and, hence, the manufacturer's decision problem in the first stage is

$$\max_{w} \pi_{Tm} = w(1 - p_T^*). \tag{3}$$

Solving (3) and combing (2), we have

**Proposition 1.** In the traditional single channel supply chain, the optimal retail price $p_T^* = (3 + c)/4$ and wholesale price $w_T^* = (1 - c)/2$. The reseller's and the manufacturer's optimal profits are given by $\pi_{Tr}^* = (1 - c)^2/16$ and $\pi_{Tm}^* = (1 - c)^2/8$.

*4.2. Decentralized Dual-Channel Supply Chain*

Now let us consider the decentralized dual-channel supply chain; the manufacturer is the Stackelberg game leader, who controls the decision of the wholesale price, and the reseller sets the retail price. To bring down the levels of channel conflict, the manufacturer adopts the same price as the reseller. The game sequence is the same as the benchmark.

By backward induction, the reseller's decision problem in the second stage is

$$\max_{p} \pi_r = (p - w - c)[s(1 - t)(1 - p) + t(1 - p)/2]. \tag{4}$$

Solving (4), we have

$$p_D^* = (1 + c + w)/2 \tag{5}$$

and, hence, the manufacturer's decision problem in the first stage is

$$\max_{w} \pi_m = w[s(1 - t)(1 - p_D^*) + t(1 - p_D^*)/2] + \\ p_D^*[(1 - s)(1 - t)(1 - p_D^*/\beta) + t(1 - p_D^*)/2]. \tag{6}$$

Solving (6) and combing (5), we obtain

**Proposition 2.** In the decentralized dual-channel supply chain, the optimal retail price $p_D^*$ and wholesale $w_D^*$ are given by

$$p_D^* = \frac{[2 + 2s(2 + c)(1 - t) + (2 + c)t]\beta}{4 + t(6\beta - 4) + 4s(1 - t)(2\beta - 1)}$$

and

$$w_D^* = \frac{[2 - t - 2c(s + t - st)]\beta - 2(1 + c)(1 - s)(1 - t)}{2 + t(3\beta - 2) + 2s(1 - t)(2\beta - 1)}, \text{ respectively.}$$

The reseller's and the manufacturer's optimal profits are denoted by $\pi_r^*$ and $\pi_m^*$ in the following. From the above analysis, the joint feasibility conditions are determined by

$$\begin{cases} p_D^* < 1, \\ w_D^* > 0. \end{cases} \tag{7}$$

Solutions of Equation (7) are given in Appendix A. While, the following discussion will be limited to the feasibility conditions. From Propositions 1 and 2, we obtain the following corollaries.

**Corollary 1.** The prices and the profits are independent of the grocery shopper's acceptance of the Internet channel but related to the Internet shopper's acceptance of the Internet channel in a decentralized dual-channel supply chain.

We find that the demand has nothing to do with the grocery shopper's acceptance of the Internet channel, so the prices and the profits will not be affected by the grocery shopper's acceptance of the Internet channel. This indicates that we make a decision without considering the grocery shopper's acceptance of the Internet channel. In Corollary 3 and the numerical examples, the impacts of the Internet shopper's acceptance of the Internet channel on the prices and the profits will be analyzed.

**Corollary 2.** (1) If $c > \frac{4(1-s)(1-t)-t}{t}$ or $\begin{cases} c < \frac{4(1-s)(1-t)-t}{t}, \\ \beta < \frac{2(3+c)(1-s)(1-t)}{4(1-s)(1-t)-(1+c)t}, \end{cases}$ then $p_D^* - p_T^* = w_D^* - w_T^* < 0$.

(2) If $\begin{cases} c < \frac{4(1-s)(1-t)-t}{t}, \\ \beta > \frac{2(3+c)(1-s)(1-t)}{4(1-s)(1-t)-(1+c)t}, \end{cases}$ then $p_D^* - p_T^* = w_D^* - w_T^* > 0$.

For proof, see Appendix B.

Corollary 2 shows that if the cost of the traditional physical channel is high enough, the retail price and the wholesale price of the dual-channel must be lower than that of traditional single channel; if the cost of the traditional physical channel is not too high, and the Internet shopper's acceptance of the Internet is not too high, the retail price and the wholesale price of the dual-channel supply chain will be lower than that of the traditional single channel, and when the Internet shoppers' acceptance of the Internet is high enough, the retail price and the wholesale price of the dual-channel will be higher than that of the traditional single channel. Furthermore, the change in the wholesale price is equal to that of the retail price.

**Corollary 3.** (1) $\frac{\partial w_D^*}{\partial \beta} = 2\frac{\partial p_D^*}{\partial \beta} > 0$.

(2) $\frac{\partial w_D^*}{\partial s} = 2\frac{\partial p_D^*}{\partial s}$; if $\beta > 1 + \frac{2+2c}{4-2t-ct}$, then $\frac{\partial w_D^*}{\partial s} < 0$; and if $\beta < 1 + \frac{2+2c}{4-2t-ct}$, then $\frac{\partial w_D^*}{\partial s} > 0$.

(3) $\frac{\partial w_D^*}{\partial t} = 2\frac{\partial p_D^*}{\partial t}$; if $\beta > \frac{(4+c)(1-s)}{3-(2-c)s}$, then $\frac{\partial w_D^*}{\partial t} < 0$; and if $\beta < \frac{(4+c)(1-s)}{3-(2-c)s}$, then $\frac{\partial w_D^*}{\partial t} > 0$.

For proof, see Appendix C.

From Corollary 3, we know: (1) The wholesale price will rise with the augmentation of the Internet shopper's acceptance of the Internet channel. (2) If the Internet shopper's acceptance of Internet channel is high, the wholesale price increases as the degree of customer loyalty to the physical channel reduces; if the Internet shopper's acceptance of the Internet channel is low, the wholesale price increases as the degree of customer loyalty to the physical channel increases. (3) If the Internet shopper's acceptance of the Internet channel is high, the wholesale price increases as the proportion of identical shoppers lowers; if the Internet shopper's acceptance of the Internet is low, the wholesale price increases as the proportion of identical shoppers increases. Meanwhile, the reseller has to adjust her retail price according to the change of the wholesale price. The results show that the retail price will increase (decrease) half as quickly as the wholesale price. As the expressions of the profits are complex, we will analyze numerically the influence of the parameters on the profits in Section 4.

### 4.3. The centralized Dual-Channel Supply Chain

In this section, we consider a centralized dual-channel supply chain in which the manufacturer controls the decision of the price $p$ from the perspective that maximizes the profit of the entire supply chain. The decision problem is

$$\max_{p} \pi_C = (p - c)[s(1 - t)(1 - p) + t(1 - p)/2] + \\ p[(1 - s)(1 - t)(1 - p/\beta) + t(1 - p)/2]. \tag{8}$$

Solving (8), we have

**Proposition 3.** In the centralized dual-channel supply chain, the optimal pricing is expressed by $p_C^* = \frac{(2 + 2cs + ct - 2cst)\beta}{4[\beta - (1 - s)(1 - t)(\beta - 1)]}$.

The profit of the centralized supply chain is denoted by $\pi_C^*$. It is not difficult to see that $0 < p_C^* < 1$ under feasible conditions. The proof is detailed in Appendix D.

From Propositions 2 and 4, we obtain the following corollary:

**Corollary 4.** The retail price in the centralized channel is lower than that in the decentralized channel, i.e., $p_C^* \leq p_D^*$; the profit in the centralized channel is higher than that in the decentralized channel, i.e., $\pi_C^* - (\pi_r^* + \pi_m^*) > 0$.

For proof, see Appendix D.

Corollary 4 shows that the decisions in a decentralized dual-channel result in channel inefficiency, so both the manufacturer and the reseller have an incentive to coordinate their decisions for a win–win strategy using the revenue distribution. The coordination is addressed in the following.

### 4.4. Coordination with A Revenue-Sharing Contract

In this section, the problem of coordinating a dual-channel supply chain will be discussed and tackled using a revenue sharing contract. We suppose that the parameter $\lambda_r(0 < \lambda_r < 1)$ represents the proportion the manufacturer receives of the revenue generated from the reseller' physical channel. Then the reseller's profits and the manufacturer's profits can be expressed by

$$\pi_{Rr} = [(1 - \lambda_r)p - w - c][s(1 - t)(1 - p) + t(1 - p)/2] \tag{9}$$

and

$$\pi_{Rm} = (\lambda_r p + w)[s(1 - t)(1 - p) + t(1 - p)/2] + \\ p[(1 - s)(1 - t)(1 - p/\beta) + t(1 - p)/2] \tag{10}$$

respectively.

The reseller's decision problem is

$$\max_{p} \pi_{Rr} = [(1 - \lambda_r)p - w - c][s(1 - t)(1 - p) + t(1 - p)/2] \tag{11}$$

Thus, $p_{Rr}^* = \frac{c + w + 1 - \lambda_r}{2(1 - \lambda_r)}$. The reseller's and the manufacturer's optimal profits under the revenue-sharing contract are denoted by $\pi_{Rr}^*$ and $\pi_{Rm}^*$.

Denote the reseller's profit margin (i.e. $\pi_{Rr}^* - \pi_r^*$) and the manufacturer's profit margin (i.e. $\pi_{Rm}^* - \pi_m^*$) by $\Delta\pi_r$ and $\Delta\pi_m$, respectively. Then

$$\Delta\pi_r = \frac{1}{32}[2s(1 - t) + t] \\ \{4 - 2\beta - t[4 - (4 - c)\beta] - 2s(1 - t)[2 - (2 - c)\beta]\}^2(\frac{1 - \lambda_r}{A^2} - \frac{4}{B^2}), \tag{12}$$

where $A = 1 - t + s(1 - t)(\beta - 1) + t\beta$, $B = 2 - t(3\beta - 2) - 2s(1 - t)(1 - 2\beta)$. As the expression of the manufacturer's profit margin is complex, and as the relationship between

the reseller's profit margin and the proportion that the manufacturer obtains of the revenue from the physical channel can show both the reseller and the manufacturer gain more profit under the revenue-sharing contract (see Proposition 4), we do not give the expression of the manufacturer's profit margin.

**Proposition 4.** (1) If the parameters $w$, $\lambda_r$ satisfy

$$w = \frac{2(1-s)(1-t)(\beta-1-c) - ct\beta + C\lambda_r}{2A}, \tag{13}$$

where $C = (2-c)t\beta + 2(1-t-\beta) - 2s(1-t)[1-(1-c)\beta]$, then the revenue-sharing contract can coordinate the dual-channel supply chain.
(2) $\frac{\partial \Delta\pi_r}{\partial \lambda_r} < 0$, $\lim\limits_{\lambda_r \to 0} \Delta\pi_r > 0$ and $\lim\limits_{\lambda_r \to 1} \Delta\pi_r < 0$.

For proof, see Appendix E.
Proposition 4 (1) indicates that it is possible to coordinate the decentralized dual-channel supply chain if the revenue-sharing contract is appropriately designed. Furthermore, Proposition 4 (2) can ensure that both the reseller and the manufacturer can obtain a win–win strategy. From this analysis, coordination of the revenue-sharing contract for a dual-channel supply chain can be implemented.

## 5. Numerical Example

### 5.1. Impact of β, s, and t on a Decentralized Dual-Channel Supply Chain

Several numerical experiments were developed to investigate the impact of the consumer acceptance, the degree of customer loyalty, and the proportion of identical shoppers on the wholesale price, the retail price, and the profits. Let $c = 0.1$. From Propositions 2 and 6, when $s = 0.3$, $t = 0.2$, $s = 0.3$, $t = 0.01$, and $s = 0.6$, $t = 0.01$, $\beta$ will vary from 1 to 3.33012. Some interesting patterns can be observed, as summarized in Figures 2–4

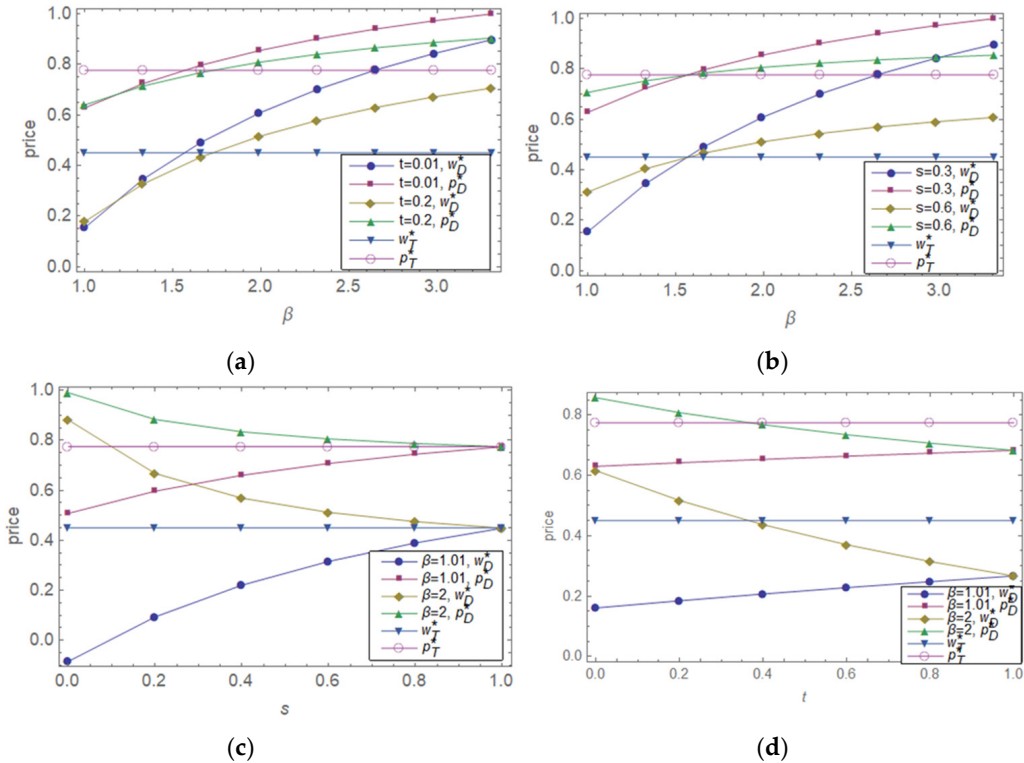

**Figure 2.** Impact of $\beta$, $s$, and $t$ on prices. (**a**) Impact of $\beta$ on price with $s = 0.3$, (**b**) Impact of $\beta$ on price with $t = 0.01$, (**c**) Impact of $s$ on price with $t = 0.01$, (**d**) Impact of $t$ on price with $s = 0.3$.

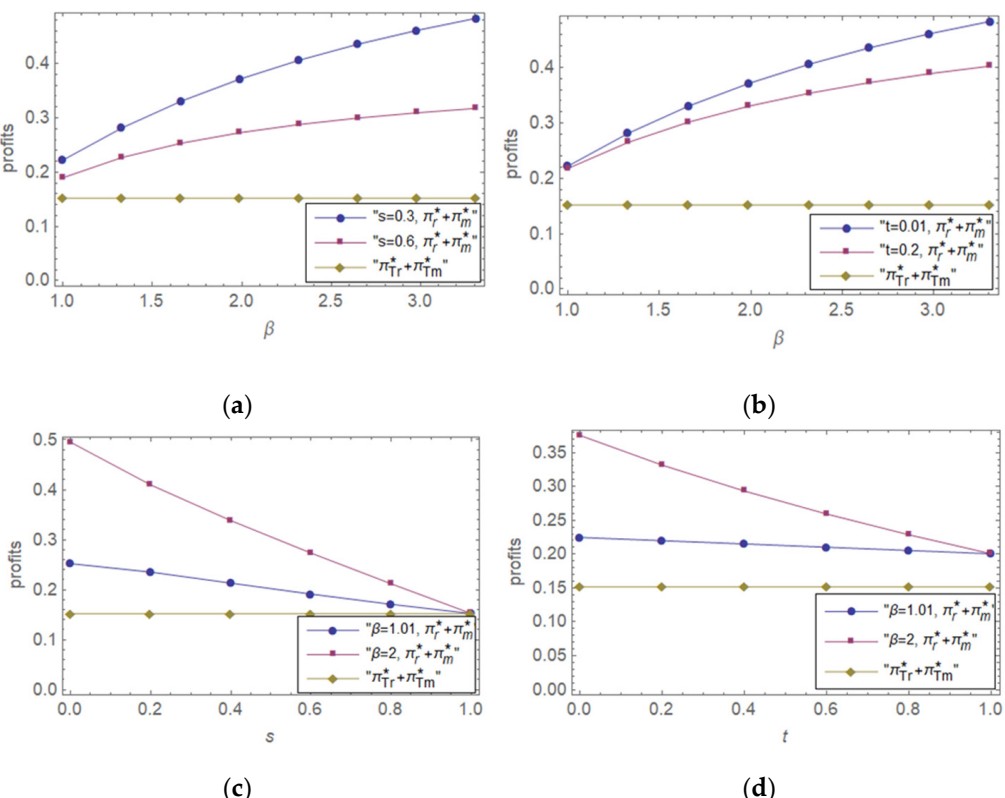

**Figure 3.** Impact of $\beta$, $s$, and $t$ on supply chain's profit. (**a**) Impact of $\beta$ on supply chain's profit with $t = 0.01$, (**b**) Impact of $\beta$ on supply chain's profit with $s = 0.3$, (**c**) Impact of $s$ on supply chain's profit with $t = 0.01$, (**d**) Impact of $t$ on supply chain's profit with $s = 0.3$.

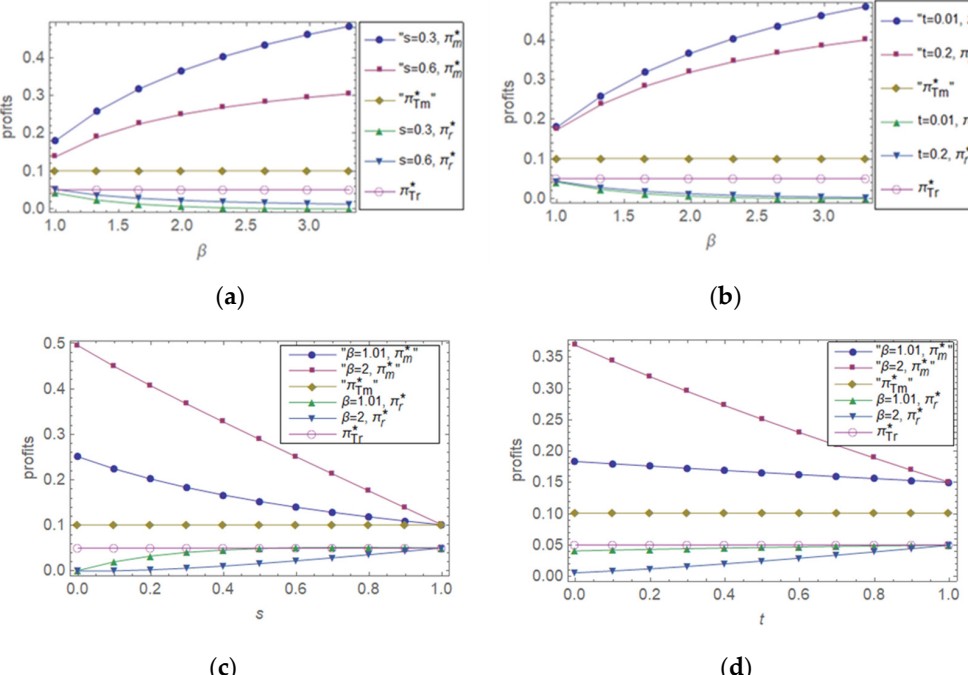

**Figure 4.** Impact of $\beta$, $s$, and $t$ on partners' profits. (**a**) Impact of $\beta$ on partners' profits with $t = 0.01$, (**b**) Impact of $\beta$ on partners' profits with $s = 0.3$, (**c**) Impact of $s$ on partners' profits with $t = 0.01$, (**d**) Impact of $t$ on partners' profits with $s = 0.3$.

### 5.1.1. Impact of $\beta$, $s$, and $t$ on Prices

From Figure 2a,b, when the proportion of identical shoppers and the degree of customer loyalty to the physical channel is fixed, the reseller prefers a higher price, as the Internet shopper's acceptance of the Internet channel increases. The wholesale price and retail price are lower than those in the traditional single channel supply chain when the Internet shopper's acceptance of the Internet is lower ($\beta$ is small), and vice versa. This is consistent with Corollary 3 (1) and Corollary 4.

Figure 2c,d show that when the Internet shopper's acceptance of the Internet channel is lower ($\beta$ is small), the wholesale price and retail price in the dual-channel supply chain will increase with the augmentation of the degree of customer loyalty (the proportion of the identical shoppers), and when the Internet shopper's acceptance of the Internet channel is higher ($\beta$ is large), the wholesale price and retail price will decrease with the augmentation of the degree of customer loyalty (the proportion of identical shoppers). This is consistent with Corollary 3 (2) and (3).

Note that $\beta$ denotes the Internet shopper's acceptance of the Internet channel; $s$ denotes the degree of customer loyalty to the physical channel; and $t$ denotes the proportions of consumers of the identical shoppers.

### 5.1.2. Impact of $\beta$, $s$, and $t$ on Supply Chain's Profits

Figure 3a,b show that the supply chain's profits will increase as the Internet shopper's acceptance of the Internet channel increases. While, the supply chain's profits will decrease with the augmentation of the degree of customer loyalty to the physical channel (the proportion of the identical shoppers), see Figure 3c,d. In any case, the supply chain's profits from the dual-channel supply chain are always larger than that of the traditional single channel supply chain. From the perspective of the supply chain, an Internet channel should be introduced in addition to the traditional retail channel.

Note that $\beta$ denotes the Internet shopper's acceptance of the Internet channel; $s$ denotes the degree of customer loyalty to the physical channel; $t$ denotes the proportions of consumers of the identical shoppers.

### 5.1.3. Impact of $\beta$, $s$, and $t$ on Partners' Profits

Figure 4 indicates that the reseller's profit is always lower than the manufacturer's profit. The reseller's profit is lower than that of the traditional single channel supply chain, and the manufacturer's profit is higher than that of the traditional single channel supply chain. From Figure 4a,b, the reseller's profit (the manufacturer's profit) will reduce (rise) with the augmentation of the Internet shopper's acceptance of the Internet channel. From Figure 4c,d, the reseller's profit (the manufacturer's profit) increases (decreases) with the augmentation of the degree of customer loyalty to the physical channel (the proportion of the identical shoppers). However, the reseller's profit can never be higher than that of the traditional single channel supply chain.

Note that $\beta$ denotes the Internet shopper's acceptance of the Internet channel; $s$ denotes the degree of customer loyalty to the physical channel; $t$ denotes the proportions of consumers of the identical shoppers.

### 5.2. The Impact of the Revenue-Sharing Contract Parameter on the Profit Margins

In this subsection, when $c = 0.1$, $t = 0.01$, $s = 0.3$, and $\beta = 2$, other parameters are kept constant, but the contract parameter $\lambda_r$ varies. To reflect the results of a win–win situation with the revenue-sharing contract (see Proposition 4), we focus on the impact of the contract parameter $\lambda_r$ on the margin of profit and then observe some interesting patterns, as summarized in Figure 5.

Figure 5 indicates that the higher the proportion the manufacturer obtains of the revenue generated from the reseller channel, the more profits the manufacturer receives. Conversely, the lower the proportion the manufacturer obtains of the revenue generated from the reseller channel, the more profits the reseller obtains. This is consistent with

Proposition 4, (2). We also find that if $\lambda_r \in [\lambda_1, \lambda_2]$, then the revenue-sharing contract can achieve a win–win strategy, signifying that this contract can be implemented.

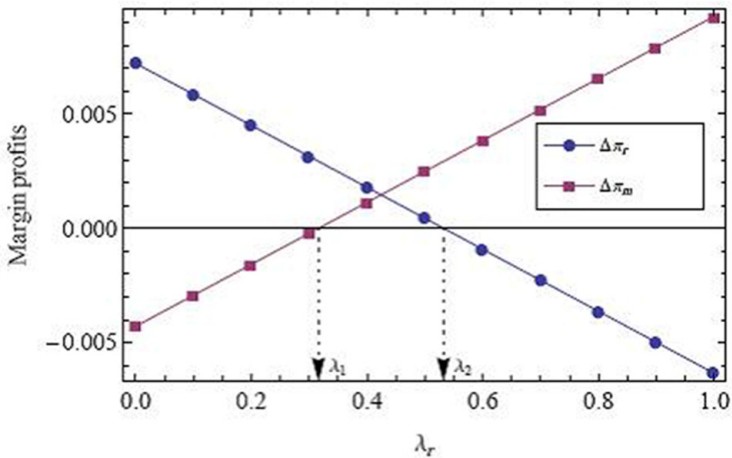

**Figure 5.** Impact of on $\lambda_r$ partners' profits.

$\lambda_r(0 < \lambda_r < 1)$ represents the proportion that the manufacturer receives of the revenue generated from the reseller channel.

## 6. Managerial Implications and Conclusion

### 6.1. Managerial Implications

In practice, when consumers can purchase the same product from an Internet channel and a physical channel, they maybe prefer to choose the channel which is suitable for their requirements or have the same willingness to buy from the Internet channel and the physical channel, especially when they can obtain the product at the same price wherever they buy it. Therefore, heterogeneous consumer channel preferences are significant in exploring supply chain pricing strategies. Our work has the following managerial implications:

First, from the perspective of the manufacturer, as the game leader, the manufacturer will increase his wholesale price with an increase of the Internet shopper's acceptance of the Internet channel. It is worth noting that, when the Internet shopper's acceptance of the Internet channel is fixed and higher, the manufacturer will raise his wholesale price as the degree of customer loyalty to the physical channel (the proportion of the identical shoppers) lowers, and vice versa. The manufacturer will always prefer to introduce dual channels and increase Internet shopper's acceptance.

Second, from the perspective of the retailer, the retailer will always prefer the traditional single channel. However, when the retailer has to face a dual-channel supply chain, she will adjust her retail price according to the manufacturer's pricing strategy and improve or decline the retail price half as quickly as the wholesale price.

Third, from the perspective of the integral supply chain, an Internet channel should be introduced in addition to the traditional retail channel, and more people should be encouraged to shop online. The supply chain can be coordinated by a designated revenue sharing contract. Although the retailer's profit is always lower than the manufacturer's, the retailer can obtain more profits when the dual-channel supply chain is coordinated. From what we have discussed, the retailer has an incentive to share profits with the manufacturer in the design of this part of the contract. Thus a win–win strategy for both partners can be implemented.

### 6.2. Conclusions

In this paper we considered a dual-channel supply chain, in which the consumers are classified into three types, according to their evaluation of the product sold by different channels. We analyze the impacts of the Internet shopper's acceptance of the Internet channel, the degree of customer loyalty to the physical channel, and the proportion of

the identical shoppers on the manufacturer's and reseller's pricing decisions and profits. The results show that the prices and the profits are independent of the grocery shopper's acceptance of the Internet channel. However, the higher the Internet shopper's acceptance of the Internet channel, the higher the prices and the supply chain's and the manufacturer's profit and the lower the reseller's profit. Meanwhile, with more consumer loyalty to the physical channel (the proportion of identical shoppers), the supply chain's profit and the manufacturer's profit will decrease and the reseller's profit will increase, but always to a level lower than that of the benchmark. Finally, we find that the revenue sharing contract can effectively coordinate a dual-channel supply chain and achieve a win-win strategy, which shows that the contract can be implemented.

Our work addresses the impact of three types of heterogeneous consumers on pricing decisions under an equal pricing strategy in a dual-channel supply chain. To the best of our knowledge, there is no research that has used a revenue-sharing contract to fully coordinate a dual-channel supply chain. Notably, this article investigates a consistent price across the reseller's physical channel and Internet channel and assumes that 50% of the identical shoppers will purchase the products from the reseller and 50% will purchase the products from the Internet channel. In practice, the price online may be higher or lower than that in the reseller's physical channel, and there may be more or less identical shoppers who go to the reseller's physical channel rather than online. In the future, extended models could be further investigated.

**Author Contributions:** R.W. and S.Y. conceived the study and completed the paper in English; S.W. participated in deriving the appendixes and drafting the article; R.W. revised it critically for important content. All authors have read and agreed to the published version of the manuscript.

**Funding:** This paper was supported by the Humanity and Social Science Youth Foundation of Ministry of Education of China( Grant No. 18YJC630274), the National Natural Science Foundation of China (Grant No. 71801085) and Doctoral Scientific Research Foundation of Henan University of Science and Technology (Grant No.13480037) and Major project of basic research on philosophy and Social Sciences in Colleges and universities of Henan Province (Research on Evaluation of Comprehensive Disaster Resilience Capacity of Community in China).

**Institutional Review Board Statement:** Not applicable.

**Informed Consent Statement:** Not applicable.

**Conflicts of Interest:** The authors declare that there is no conflict of interest regarding the publication of this paper.

## Appendix A. Solutions of (7)

Note that $p_D^* < 1$ is equivalent to

$$4(1-s)(1-t) + \beta D > 0, \tag{A1}$$

where $D = 4s(1-t) + 4t - 2 - c[2s(1-t) + t]$. It follows from (A1) that

$$p_D^* < 1 \Leftrightarrow c < \frac{4s(1-t) + 4t - 2}{2s(1-t) + t} \text{ or } \begin{cases} c > \frac{4s(1-t)+4t-2}{2s(1-t)+t}, \\ 1 < \beta < \frac{4(1-s)(1-t)}{c[2s(1-t)+t]-[2-4s(1-t)-4t]}. \end{cases} \tag{A2}$$

By a direct calculation, we know

$$w_D^* > 0 \Leftrightarrow 2(1+c)(1-s)(1-t) + [t + 2c(s+t-st) - 2]\beta < 0. \tag{A3}$$

Thus,

$$w_D^* > 0 \Leftrightarrow \begin{cases} c < \frac{2-t}{2(s+t-st)}, \\ \beta > \frac{2(1+c)(1-s)(1-t)}{2-t-2c(s+t-st)}. \end{cases} \tag{A4}$$

From the above, we get the solutions of (7) as the following

When $\frac{(1-s)(1-t)}{t} > \frac{1}{2}$,

$$
\begin{cases}
\frac{2-t}{2(s+t-st)} > c > \frac{4s(1-t)+4t-2}{2s(1-t)+t}, \\
max\{1, M_1\} < \beta < M_2,
\end{cases}
\tag{A5}
$$

or

$$
\begin{cases}
c < \frac{4s(1-t)+4t-2}{2s(1-t)+t}, \\
max\{1, M_1\} < \beta,
\end{cases}
\tag{A6}
$$

and when $\frac{(1-s)(1-t)}{t} < \frac{1}{2}$,

$$
\begin{cases}
c < \frac{2-t}{2(s+t-st)}, \\
max\{1, M_1\} < \beta,
\end{cases}
\tag{A7}
$$

where $M_1 = \frac{2(1+c)(1-s)(1-t)}{2-t-2c(s+t-st)}$, $M_2 = \frac{4(1-s)(1-t)}{c[2s(1-t)+t]+[2-4s(1-t)-4t]}$.

## Appendix B. Proof of Corollary 2

By a direct calculation, we know

$$
p_D^* - p_T^* = w_D^* - w_T^* = \frac{[4(1-s)(1-t)-t-ct]\beta - 2(3+c)(1-s)(1-t)}{8(1-s)(1-t)+4(4s+3t-4st)\beta}
\tag{A8}
$$

Thus if $c > \frac{4(1-s)(1-t)-t}{t}$, then $p_D^* - p_T^* = w_D^* - w_T^* < 0$; if $\begin{cases} c < \frac{4(1-s)(1-t)-t}{t} \\ \beta < \frac{2(3+c)(1-s)(1-t)}{4(1-s)(1-t)-(1+c)t} \end{cases}$,

then $p_D^* - p_T^* = w_D^* - w_T^* < 0$; and if $\begin{cases} c < \frac{4(1-s)(1-t)-t}{t} \\ \beta > \frac{2(3+c)(1-s)(1-t)}{4(1-s)(1-t)-(1+c)t} \end{cases}$, then

$p_D^* - p_T^* = w_D^* - w_T^* > 0$.

## Appendix C. Proof of Corollary 3

(1) It is easy to show that

$$
\frac{\partial w_D^*}{\partial \beta} = 2\frac{\partial p_D^*}{\partial \beta} = \frac{2(1-s)(1-t)[2+2s(2+c)(1-t)-t(2+c)]}{[2+t(3\beta-2)+2s(1-t)(2\beta-1)]^2} > 0
\tag{A9}
$$

(2) A direct calculation shows that

$$
\frac{\partial w_D^*}{\partial s} = 2\frac{\partial p_D^*}{\partial s} = \frac{2(1-t)[6+c(2+t\beta-t)+2t(\beta-1)-4\beta]\beta}{[2+t(3\beta-2)+2s(1-t)(2\beta-1)]^2}
\tag{A10}
$$

Thus, $\frac{\partial w_D^*}{\partial s} > 0 \Leftrightarrow \beta < 1 + \frac{2+2c}{4-2t-ct}$ and $\frac{\partial w_D^*}{\partial s} < 0 \Leftrightarrow \beta > 1 + \frac{2+2c}{4-2t-ct}$.

(3) A direct calculation shows that

$$
\frac{\partial w_D^*}{\partial t} = 2\frac{\partial p_D^*}{\partial t} = \frac{2\beta[(4+c)(1-s)-(3-2s+cs)\beta]}{[2+t(3\beta-2)+2s(1-t)(2\beta-1)]^2}
\tag{A11}
$$

Thus, $\frac{\partial w_D^*}{\partial t} < 0 \Leftrightarrow \beta > \frac{(4+c)(1-s)}{3+(-2+c)s}$ and $\frac{\partial w_D^*}{\partial t} > 0 \Leftrightarrow \beta > \frac{(4+c)(1-s)}{3+(-2+c)s}$.

## Appendix D. Proof of Proposition 3 and Corollary 4

A direct calculation shows that

$$
p_C^* - p_D^* = \frac{\beta(2st-2s-t)(M_3-M_4\beta)}{4M_5 M_6}
\tag{A12}
$$

where $M_3 = 4(1-s)(1-t)$, $M_4 = 2(1-s)(1-t) - 2 + 2cs + ct + 4st - 2cst$, $M_5 = -\beta s + \beta st - st + s - \beta t + t - 1$, $M_6 = 4\beta s - 4\beta st + 3\beta t + 2(1-s)(1-t)$.

Combining Appendix A, one can show in any case of the joint feasibility conditions $0 < p_C^* \leq p_D^* < 1$.

Additionally,

$$\pi_C^* - (\pi_r^* + \pi_m^*) = \frac{2s(1-t) + t^2 \beta M_7^2}{16 A M_8^2} > 0 \tag{A13}$$

where $A = 1 - t + s(1-t)(\beta - 1) + t\beta$, $M_7 = 4 - 2\beta - t[4 - (4-c)\beta] - 2s(1-t)[2 - (2-c)\beta]$, $M_8 = 2 + t(3\beta - 2) - 2s(1-t)(2\beta - 1)$.

## Appendix E. Proof of Proposition 4

(1)   From the coordination condition of the dual-channel supply chain, we have

$$p_{Rr}^* = p_C^* \tag{A14}$$

and then

$$w = \frac{2(1-s)(1-t)(\beta - 1 - c) - ct\beta + C\lambda_r}{2A} \tag{A15}$$

(2)   It is easy to show that

$$\frac{\partial \Delta \pi_r}{\partial \lambda_r} = -\frac{[2s(1-t) + t]M_7^2}{32A^2} < 0 \tag{A16}$$

and

$$\lim_{\lambda_r \to 0} \Delta \pi_r = \frac{1}{32}[2s(1-t) + t]M_7^2 \frac{(2s + t - 2st)\beta M_9^2}{A^2 M_{10}^2} > 0, \tag{A17}$$

$$\lim_{\lambda_r \to 1} \Delta \pi_r = -\frac{[2s(1-t) + t]M_7^2}{8M_8^2} < 0, \tag{A18}$$

where $A$, $M_7$, $M_8$ are the same as in Appendix D and $M_9 = 4(1-s)(1-t) + 6s\beta + 5t\beta - 6st\beta$, $M_{10} = 2(1-s)(1-t) + 4s\beta + 3t\beta - 4st\beta$, $C = (2-c)t\beta + 2(1-t-\beta) - 2s(1-t)[1 - (1-c)\beta]$.

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
