# Peer review of "Pricing and Coordination Strategies of Dual Channels Considering Consumers’ Channel Preferences"

_sustainability, doi:10.3390/su132011191_

Round 1

Reviewer 1 Report

The paper is about price strategies in a dual-channel supply chain. The thematic is very interesting and relevant. I am found the introduction/ review section very clear and informative. However, I have many problems understanding the methods and results. Maybe the difficulties concern my personal limitations; however, many other readers could face the same problem. The authors could revise these sections by including more information to help the reader, especially those not used to with econometrics.

Abstract:

L10 – remove “channel” word; it is duplicate

It is hard to understand the abstract; I think you should include more information about the method. Is it a simulation study? Which method and theoretical background were used?

You should include a conclusion section in the abstract.

Introduction – The introduction is very clear!

L36 – In this case, is dual-channel without a hyphen? Please be consistent.

Literature review- I think you should include some words about the Stackelberg leadership model

Methods – I had some difficulties understanding the methods. You focused on describing the equations clearly, but it is not clear how this clearly simulates the real world. Also, I haven’t received the appendix

L130 – Is it a simulation study?

Results

The figures are very hard to understand. We have to return to the text to see what beta, s, t, w, p etc are. Figures must stand by themselves.

I am not used to this format of paper. Where is the discussion section? Where are the practical implications and limitations? I think you could discuss your results in a practical way, helping practitioners and researchers from other areas such as consumer’s behavior.

Author Response

1
Reply to Reviewer
1
Thank you for reviewing our paper entitled “
Pricing and Coordination
Strategies of Dual channels Considering Consumers’ Channel Preferences We
are grateful for the feedback of you, which has led to an improvement of our paper.
We list the comments of you below (in black regular font), followed by our responses
(in blue regular font).
Reviewer #
1 : The paper is about price strategies in a dual channel supply chain.
The thematic is very interesting and relevant. I am found the introduction/ review
section very clear and informative. However, I have many problems understanding the
methods and results. Maybe the difficulties concern my personal lim itations; however,
many other readers could face the same problem. The authors could revise these
sections by including more information to help the reader, especially those not used to
with econometrics.
Thank you for your careful reading our paper and
constructive suggestions and
comments. We especially thank you for giving us the opportunity to enhance the paper.
We have undertaken a very serious revision according to all your suggestions.
Highlights of the changes to your comments are as follows:
1. We h ave added some information about the method.
2. I n Literature review, we have added some words to interpret the Stackelberg
leadership model
3. We have also included a new section denoted as “Discussions and Managerial
implications” to further discuss the managerial implications and impacts for
real world practices explicitly.
At the same time, the detailed responses with respect to each comment are
provided below.
2
Abstract:
Abstract:
L
L10 10 –– remove “channel” word; it is duplicateremove “channel” word; it is duplicate..
Response: Thank you very
Response: Thank you very muchmuch for your careful reading and pointing out the for your careful reading and pointing out the error error and we have removed and we have removed the the “channel” word“channel” word. . Meanwhile, wMeanwhile, we have corrected all the e have corrected all the similar errors word by word in the revision. The writing and presentation have been similar errors word by word in the revision. The writing and presentation have been improved in our best knowledge.improved in our best knowledge.
It is har
It is hard to understand the d to understand the abstractabstract; I think you should include more information ; I think you should include more information about the method. about the method.
Response: Thank
Response: Thanks a lot. s a lot. We have We have added some information to introduce the added some information to introduce the Stackelberg Stackelberg gamegame modelmodel::
A
A Stackelberg game assumeStackelberg game assumess that that tthe he manufacturer manufacturer as the game leaderas the game leader makemakess wholesale price firstlywholesale price firstly..Then tThen the he reresellerseller decides the retail price accordingdecides the retail price according toto the the wholesale price andwholesale price and sellsellss the product the product to the consumersto the consumers..
Is it a simulation study?
Is it a simulation study?
Response: Thanks
Response: Thanks.. TThishis study is not a simulationstudy is not a simulation, , wewe firstfirst estestablishablish Stackelberg Stackelberg game game modelmodelss then mthen make a theoretical analysisake a theoretical analysis using mathematics methodusing mathematics method about the model. about the model. Furthermore, Furthermore, in order to intuitionally observe the influences of parameters on the in order to intuitionally observe the influences of parameters on the supply chain, we develop some the numericalsupply chain, we develop some the numerical experiments.experiments.
Which method and
Which method and theoretical background were used?theoretical background were used?
Response:
Response:ThanksThanks.. WWe use game theory to establish our models and de use game theory to establish our models and derive erive thethe equilibrium solutionequilibrium solutions bs by backward inductiony backward induction..
You should include a conclusion section in the abstract.
You should include a conclusion section in the abstract.
Response:
Response:Thanks for your reminding.Thanks for your reminding. According to your suggestions, we add a According to your suggestions, we add a conclusionconclusion iin the last sentencen the last sentence. In addition, we . In addition, we have rewritten the abstract and added have rewritten the abstract and added the effect of the parameters on prices and profits discussed in the articlethe effect of the parameters on prices and profits discussed in the article..
3
Introduction
Introduction –– The introduction is very clear!The introduction is very clear!
Re
Response:sponse:TThank youhank you very muchvery much for your appreciating our work.for your appreciating our work.
L36
L36 –– In this case, is dualIn this case, is dual--channel without a hyphen? Please be consistent.channel without a hyphen? Please be consistent.
Response:Thank you very much for your careful reading and pointing out that the
Response:Thank you very much for your careful reading and pointing out that the format should be consistent. We have format should be consistent. We have adopted “dualadopted “dual--channel”channel” throughout the article throughout the article and checked the format word by word in the revision.and checked the format word by word in the revision.
Literature review
Literature review-- I think you should include some words about the Stackelberg I think you should include some words about the Stackelberg leadership modelleadership model..
Response:Thank you
Response:Thank you for your constructive suggests. for your constructive suggests. WeWe add some information about add some information about the the Stackelberg leadership modelStackelberg leadership model and and hope hope the information can help the the information can help the readers readers understand our paper better. Tunderstand our paper better. Thehe information that we have added is writteninformation that we have added is written as follows:as follows:
Different from the Nash bargaining game in
Different from the Nash bargaining game in Li et al. [Li et al. [66], we ], we establish a establish a Stackelberg Stackelberg game model where the manufacturer is the game leader and the retailer is the game game model where the manufacturer is the game leader and the retailer is the game follower.follower. Stackelberg gameStackelberg game was proposed by was proposed by Heinrich Von StackelbergHeinrich Von Stackelberg who was a who was a German economistGerman economist in in StackelbergStackelberg (1952)(1952)[13][13].. As well known, in a As well known, in a Stackelberg game Stackelberg game one player acts first and the other player acts laterone player acts first and the other player acts later.. MMany scholars use any scholars use Stackelberg Stackelberg gamegamess to investigate to investigate price strategiesprice strategies in supply chain managementin supply chain management [[14,15,1614,15,16]]. As the . As the same as same as Li et al. [Li et al. [66]], we adopt, we adopt a consistent pricinga consistent pricing inin the dualthe dual--channel channel supply chain.supply chain.
Methods
Methods –– I had some difficulties understanding the methods. You focused on I had some difficulties understanding the methods. You focused on describing the equations clearly, but it is not clear how this clearly simulates the real describing the equations clearly, but it is not clear how this clearly simulates the real world. world.
Response:Thank you.
Response:Thank you. To help the readers understanTo help the readers understand d Stackelberg gameStackelberg game method better, method better, we give we give tthe game sequence as follows (as shown in Fig. he game sequence as follows (as shown in Fig. 11).).
4
Fig. 1: The game sequence
Fig. 1: The game sequence
Also, I haven’t received the appendix
Also, I haven’t received the appendix..
Response:Thanks for your reminding.
Response:Thanks for your reminding. In the earlier In the earlier version version of this manuscriptof this manuscript, we , we submitted the appendixsubmitted the appendix inin a Latex format. a Latex format. Maybe the system only Maybe the system only identifies the identifies the uunified nified template.template. In the new revisionIn the new revision MSMS wordword,, we have added the section we have added the section “A“Appendixppendix”. ”.
L130
L130 –– Is it a simulation study?Is it a simulation study?
Response:Thank you.
Response:Thank you. We have answeWe have answered this question in the above.red this question in the above.
Results
Results
The figures are very hard to understand. We have to return to the text to see what beta,
The figures are very hard to understand. We have to return to the text to see what beta, s, t, w, p etc are. Figures must stand by themselves.s, t, w, p etc are. Figures must stand by themselves.
Response:Thanks for your reminding.
Response:Thanks for your reminding. In the new revision, tIn the new revision, the meaninghe meaningss of of each each parameterparameter has beenhas been explained below explained below every figure. The added writing is every figure. The added writing is marked up marked up using the “Track Changes” functionusing the “Track Changes” function. .
I am not used to this format of paper. Where is the discussion section? Where are the
I am not used to this format of paper. Where is the discussion section? Where are the practical implications and limitations? I tpractical implications and limitations? I think you could discuss your results in a hink you could discuss your results in a practical way, helping practitioners and researchers from other areas such as practical way, helping practitioners and researchers from other areas such as consumer’s behavior.consumer’s behavior.
Response:
Response: Thanks a lot. Thanks a lot. We have included a new section denoted as “We have included a new section denoted as “Discussions and Discussions and
First stage
First stage::the manufacturerthe manufacturer
decides the wholesale price decides the wholesale price ?
Second stage
Second stage::the retailer decides the retail price the retailer decides the retail price ?
Third stage
Third stage::the consumer the consumer decides decides which channel to purchase the product which channel to purchase the product fromfrom which channel.which channel.
5
managerial implications
managerial implications” before Conclusion to discuss the managerial implications ” before Conclusion to discuss the managerial implications and impact for real world practices explicitly. In the revision, these are and impact for real world practices explicitly. In the revision, these are marked up marked up using the “Track Changes” functusing the “Track Changes” function inion in Section 6.Section 6.
In practice, when the consumers can purchase the same product
In practice, when the consumers can purchase the same product from Internet from Internet channel and the physical channel, they maybe prefer to choose the channel which is channel and the physical channel, they maybe prefer to choose the channel which is suitable for their requirement or have the same willingness to pay on Internet channel suitable for their requirement or have the same willingness to pay on Internet channel and the physical channel, especially when they can obtain the product atand the physical channel, especially when they can obtain the product at the same the same price wherever they buy. So heterogeneous consumers’ channel preference is price wherever they buy. So heterogeneous consumers’ channel preference is significant in exploring supply chain pricing strategies. Our work has the following significant in exploring supply chain pricing strategies. Our work has the following managerial implications. managerial implications.
First, from the perspective of the manufacturer, as the game
First, from the perspective of the manufacturer, as the game leader, the leader, the manufacturer will rise up his wholesale price with the augment of the Internet manufacturer will rise up his wholesale price with the augment of the Internet shopper’s acceptance of Internet channel. It’s worth noting that when the Internet shopper’s acceptance of Internet channel. It’s worth noting that when the Internet shopper’s acceptance of Internet channel is fixed and higher, the manufacturer wishopper’s acceptance of Internet channel is fixed and higher, the manufacturer will ll rise up his wholesale price as the degree of customer loyalty to the physical channel rise up his wholesale price as the degree of customer loyalty to the physical channel (the proportion of the identical shoppers) slows down, and vice versa. The (the proportion of the identical shoppers) slows down, and vice versa. The manufacturer will always prefer to introduce the dual channels and higher Internet manufacturer will always prefer to introduce the dual channels and higher Internet shopper’sshopper’s acceptance. acceptance.
Second, from the perspective of the retailer, the retailer will always prefer to the
Second, from the perspective of the retailer, the retailer will always prefer to the traditional single channel. However, when the leaded retailer has to face the traditional single channel. However, when the leaded retailer has to face the dualdual--channel supply chain, she will adjust her retail price according to the channel supply chain, she will adjust her retail price according to the mamanufacturer’s pricing strategy and improve or decline her retail price half as quickly nufacturer’s pricing strategy and improve or decline her retail price half as quickly as the wholesale price.as the wholesale price.
Third, from the perspective of the integral supply chain, Internet channel should
Third, from the perspective of the integral supply chain, Internet channel should be introduced in addition to the traditional retail channel and be introduced in addition to the traditional retail channel and more people should be more people should be encouraged to go to shop online. The supply chain can be coordinated by a designed encouraged to go to shop online. The supply chain can be coordinated by a designed revenue sharing contract. Although the retailer’s profit is always lower than the revenue sharing contract. Although the retailer’s profit is always lower than the manufacturer’s, the retailer can obtain more profits when the dualmanufacturer’s, the retailer can obtain more profits when the dual--chchannel supply annel supply chain is coordinated. From what we have discussed, the retailer has incentive to share chain is coordinated. From what we have discussed, the retailer has incentive to share profits with the manufacturer in the design of this part of the contract. Thus a winprofits with the manufacturer in the design of this part of the contract. Thus a win--win win
6
strategy for both partners can be implemented.
strategy for both partners can be implemented.

Reviewer 2 Report

The authors assessed the impact of three types of heterogeneous consumers on pricing decisions under equal pricing strategy in a dual-channel supply chain. The paper is clearly written and well structured. I only have two comments:

1- The authors should extend the discussion of their results. For example, statements like "prices and the profits are independent of the grocery shopper’s acceptance of the Internet channel." should be well discussed. Some of the results are noted in line with the literature, so, they should be well discussed. 

2- I suggest to add a section on the limitations of the study. 

Thank you,

Best.   

Author Response

1
Reply to Reviewer 2
Thank you for reviewing our paper entitled “Pricing and Coordination
Strategies of Dual channels Considering Consumers’ Channel Preferences”. We
are grateful for the feedback of you, which has led to an improvement of our paper.
We list the comments of you below (in black regular font), followed by our responses
(in blue regular font).
Reviewer#2:The authors assessed the impact of three types of heterogeneous
consumers on pricing decisions under equal pricing strategy in a dual-channel supply
chain. The paper is clearly written and well structured. I only have two comments:
Thank you for your careful reading our paper and constructive suggestions and
comments. We especially thank you for giving us the opportunity to enhance the paper.
We have undertaken a very serious revision according to all your suggestions.
Highlights of the changes to your comments are as follows:
1. We have included a new section denoted as “Discussions and Managerial
implications” to further discuss the managerial implications of the results for
real world practices explicitly.
2. We have added some limitations of the study in the end.
At the same time, the detailed responses with respect to each comment are
provided below.
1- The authors should extend the discussion of their results. For example, statements
like "prices and the profits are independent of the grocery shopper’s acceptance of the
Internet channel." should be well discussed. Some of the results are noted in line with
the literature, so, they should be well discussed.
Response: Thanks a lot. We have discussed “prices and the profits are independent of
the grocery shopper’s acceptance of the Internet channel.” and are marked up using
the “Track Changes” function in Corollary 1. Addtionally, we have included a new
section denoted as “Discussions and managerial implications” before Conclusion to
discuss the managerial implications and impact for real world practices explicitly.
2
2- I suggest to add a section on the limitations of the study.
Response:Thanks for your reminding. In the new revision, we have added some
limitations of our study. The added limitations are written as follows and are marked
up using the “Track Changes” function in the end.
Note that this article investigates consistent price across the reseller’s physical
channel and Internet channel and assumes that 50% of the identical shoppers will
purchase the products from the reseller and 50% will purchase the products from
Internet channel. In practice, the price online may be higher or lower than that in the
reseller’s physical channel and there may be more or less identical shoppers that go to
the reseller’s physical channel than online. In the future, the extended models could
be further investigated.

Round 2

Reviewer 1 Report

The manuscript is much improved. I think it could be accepted in the present form.